# Evaluating route preview as an alternative to turn-by-turn navigation in pedestrian mobility

**Gian-Luca Savino[1,2], Emanuel de Bellis[1], Reuben Kirkham[3,4], Johannes Schöning[1]***

**1** University of St. Gallen, St. Gallen, Switzerland, **2** Gottlieb Duttweiler Institute (GDI), Rüschlikon, Switzerland, **3** Monash University, Clayton, Victoria, Australia, **4** Free Speech Union, Melbourne, Australia

\* johannes.schoening@unisg.ch

## Abstract

Mobile navigation apps like Google Maps and Apple Maps predominantly use turn-by-turn (TBT) instructions for pedestrians, a paradigm originally designed for vehicular navigation. However, many users actively choose the simpler route preview (RP) mode, which displays only the mapped route without real-time guidance features. This research employs a three-study mixed-methods approach to investigate whether RP could serve as an effective alternative to TBT for pedestrian navigation. Study 1 surveyed user preferences (n = 222), revealing that 44% prefer RP despite its limited features, rising to 76% in familiar environments. Study 2 compared actual navigation performance (n = 195), finding no significant differences between modes across key metrics: navigation errors, phone glances, and spatial learning, with RP on par and even better for some key metrics. Study 3 was a co-design workshop (n = 5) to identify priority enhancements, including landmark integration, intention-based routing, and subtle orientation aids. Overall, these findings challenge the assumption that TBT represents optimal pedestrian navigation design. Instead, they demonstrate that RP, with its cognitive engagement benefits and user autonomy, performs comparably to TBT despite fewer features. When enhanced with targeted improvements, RP could better serve diverse pedestrian needs than current TBT-dominant interfaces. This work provides evidence-based guidance for developing more versatile, cognitively engaging pedestrian navigation systems.

## 1 Introduction

Google Maps and Apple Maps have become indispensable tools for navigating our cities, used by millions of pedestrians daily. These applications offer turn-by-turn (TBT) navigation as their default mode—providing step-by-step directions with voice guidance to lead users along their routes. Yet surprisingly, many users ignore these sophisticated features and instead use the simpler route preview (RP) mode, which merely displays the path on a map without any active guidance. This widespread

**Data availability statement:** All relevant data are within the paper and its Supporting Information files.

**Funding:** GS and JS - Swiss National Science Foundation (SNSF) grant number 200021 207430.

**Competing interests:** The authors have declared that no competing interests exist.

behaviour raises a fundamental question: Is TBT navigation, initially designed for driving, actually the better approach for pedestrian wayfinding?

Current mobile navigation apps follow a two-step process: first, calculating and displaying routes as a preview on the map (sometimes with alternatives), then providing TBT instructions with auditory or multimodal feedback. While this approach works well for vehicular navigation, where drivers must follow fixed roads, pedestrians have fundamentally different needs and freedoms. Recent research increasingly questions TBT's suitability for walking.

Existing research highlights the limitations of current turn-by-turn (TBT) navigation systems in pedestrian contexts. Pielot et al. [1] emphasise that these systems prioritise speed and direction over exploration, while Rousell and Zipf [2] note their limited adaptation to pedestrian preferences, such as landmarks. Basiri et al. [3] further argue that TBT systems inadequately account for pedestrians' greater movement freedom. Supporting this, Savino et al. [4] show that Google Maps users spend most of their time scrolling through the map, suggesting a desire for exploration and autonomy, which positions route previews (RP) as a promising and simple alternative to guided TBT navigation.

Extending these observations, Toups Dugas et al. [5] analyse mobile maps through reflective auto-ethnography, identifying limitations such as poor vertical representation, missing local detail, inaccurate sensor data, and ineffective path generation. Similarly, Mazurkiewicz et al. [6] note that navigation assistance systems may impair spatial cognition due to the divided attention required. They propose *Free Choice Navigation*, which balances user decisions, route length, and instructions, and show through agent-based simulation that users can make independent choices at most junctions while reliably reaching their destinations. Together, these studies emphasise the need for navigation systems that support exploration, autonomy, and engagement with the environment.

Navigating extensive and unfamiliar virtual environments poses considerable challenges. Vinson [7] proposes a set of design guidelines intended to enhance navigational performance within such environments. The recommendations focus primarily on the design and strategic placement of landmarks. These guidelines are grounded in extensive empirical research on navigation in real-world contexts, reflecting the assumption that human navigational behaviour exhibits significant parallels across virtual and physical spaces.

Still, the RP lacks certain features that are present when using TBT navigation across many mobile navigation applications. For example, TBT automatically rotates the map so that it always faces forward, which helps users align themselves with the map and reduces the mental effort needed to navigate [8].

Another feature missing in current implementations of RP is adequate feedback from the user interface. The user's location indicator moves, but the rest of the interface does not update to reflect the progress made on the route, which conflicts with standard user interface design principles [9]. Estimated times of arrival (ETAs) also remain static, as the RP mode appears to be designed primarily for initial route planning rather than active navigation.



Based on the identified gaps in pedestrian navigation literature, we formulate three specific research questions:

- RQ1: What are the stated preferences of pedestrians regarding route preview (RP) versus turn-by-turn (TBT) navigation modes, and how do contextual factors influence these preferences?

- RQ2: How do RP and TBT navigation modes compare in terms of objective performance metrics (navigation errors, attention demands, spatial learning) when used by pedestrians?

- RQ3: What design enhancements could improve RP navigation to better serve pedestrian needs while maintaining its cognitive benefits?

These questions address the gap between the theoretical advantages of survey knowledge (promoted by RP) and the practical dominance of TBT in current pedestrian navigation systems. While TBT navigation remains the prevailing approach in practice, it is important to consider that RP navigation may offer distinct cognitive benefits.

This increased cognitive activity likely leads to better spatial learning outcomes [10]. Similarly, Bakdash et al. [11] demonstrate that active decision-making enhances environmental engagement and spatial knowledge acquisition. Thus, RP as a navigation mode facilitates richer cognitive engagement, better spatial understanding, and greater flexibility in navigating around potential barriers or following personal preferences.

Previous research suggests that TBT instructions may not be the most suitable navigation mode for pedestrians, and that they may therefore actively use alternatives that better suit their personal needs [1,5,6,12,13]. To test this hypothesis, we employed a three-stage study design. In this paper, we present the results of a representative online survey (n = 222), a navigation field study (n = 195), and a design studio workshop (n = 5).

Through our online survey (n = 222) in Study 1, we found that a surprising 44% of users stated that they prefer using the RP mode for navigation, despite it not being a primary feature designed for this purpose in mobile navigation apps. For familiar environments, this preference was even more pronounced, with 76% of users opting for RP. While the RP merely provides an overview of the route and allows users to choose the desired route options, many users prefer this feature for their active navigation tasks, ignoring the TBT navigation feature entirely.

We conducted a navigation study (study 2) to investigate navigation performance under both modes. We measured navigation errors, the times participants looked at their phones while running a mobile navigation application, and participants' ability to correctly draw the route after arriving at the predefined destination of their route. The results we obtained are surprising. Despite the widespread implementation of TBT in major mobile navigation applications, we found no significant difference between the two modes in terms of navigation performance, attention during navigation, and spatial learning.

However, we found that, across both modes, participants who were more unfamiliar with the environment had to look at their phones more often. Participants who use mobile navigation apps more regularly were better at indicating the correct turn directions during the route sketching task.

Based on this result, we see potential for improving the design of current RP implementations. With the incorporation of additional features currently present in the TBT navigation method, we argue that RP navigation has the potential to enhance the user experience and conceivably surpass TBT navigation in performance. Therefore, in the last step, we conducted a design studio workshop (study 3). During the workshop, participants envisioned and designed critical features to enhance the usefulness of the RP mode. Among the most favoured features were the incorporation of landmarks and ETA/progress updates into the RP navigation mode.

By recognising and making use of the unique advantages offered by RP, designers and developers are presented with an opportunity to design mobile navigation applications that are more versatile, adaptable, and tailored to the needs of pedestrians, rather than being clones of what is used for in-car navigation these days.



The remainder of this article is organised as follows. Section 2 reviews related work examining how humans, and pedestrians in particular, navigate through environments and the types of information they use for wayfinding and orientation. This review informs the identification of relevant features for the further development of Route Preview (RP) navigation for pedestrians. Section 3.2 presents the results of Study 1, an online survey designed to explore individuals' stated preferences for either RP or Turn-by-Turn (TBT) navigation modes. Section 3.3 reports Study 2, a field experiment that investigates actual navigation performance differences between RP and TBT. Section 3.4 describes Study 3, a design studio workshop conducted to identify and develop critical features to enhance the usefulness and potential of the RP mode. Section 4 presents the overall findings, with a focus on the role of RP and its design implications. Finally, Section 5 provides the conclusion.

## 2  Literature review

In this section, we present research on how humans, and pedestrians in particular, navigate through known and unknown environments, as well as the information they use to plan and orient themselves during this process. This foundational understanding will help us identify relevant features to further develop the RP navigation for pedestrians.

### 2.1  Wayfinding strategies

When humans attempt to navigate an environment, they utilise elements from that environment to orient themselves. Famously, Lynch [14] outlined five urban elements – edges, paths, landmarks, districts, and nodes – that aid in creating a mental image of a city, with landmarks playing the most important role in spatial knowledge acquisition and wayfinding [15], as navigators use visible landmarks to orient themselves [16,17]. While there are various definitions of landmarks, the most relevant characteristics during navigation are visual importance, historic or cultural importance, and structural importance [18]. Furthermore, landmarks should be permanent, visible, unique and in the right location in relation to the user [19]. Landmarks that fulfil these criteria can aid in wayfinding, especially if no digital aid like mobile navigation applications are used. Beyond using landmarks to help with navigation, there is also route-based navigation, which involves remembering the sequence of paths, including turns and intersections, between landmarks [16]. Route knowledge is quickly obtained even after walking a route just once without any other distractions [20]. While this can be very efficient, using guided navigation through modern mobile navigation applications [4,12,21], people acquire less route knowledge than participants with low or non-automated navigation tools [22]. Lastly, there is survey knowledge. Survey knowledge relies on a cognitive map of the environmental layout, often acquired by a birds-eye view map, allowing navigators to understand the relationship between urban elements [23,24]. This type of knowledge allows individuals to construct cognitive maps, which are essential for understanding the configuration of streets, landmarks, and other key features within a city [25]. By facilitating an overarching view of the environment, survey knowledge empowers pedestrians to navigate more flexibly, making it easier to plan routes and adapt to changes without relying on TBT directions [26]. Interactive maps that allow users to explore the area by zooming and panning can help develop these cognitive maps, enabling users to understand the spatial layout and anticipate the position of landmarks [27]. This is particularly beneficial in unfamiliar environments, where survey knowledge supports better orientation and decision making [25].

Naturally, these wayfinding strategies do not appear in isolation, but rather in combination with each other, as people switch and combine strategies during a wayfinding task [28]. This means that a good navigation application should be based on all three strategies: landmark-knowledge, route-knowledge, and survey-knowledge combined to offer the best user experience. Nowadays, no navigation mode for pedestrians fully implements all three concepts. While in some modes, survey knowledge is more promoted (e.g., RP navigation), others depend fully on route-based navigation (e.g., TBT navigation), and landmarks are only rarely used during active navigation in such systems.

## 2.2 Map rotation

Traditional maps (e.g., paper maps) are designed so north is up [29]. This convention provides a consistent reference point, aiding in the standardisation of maps as research has found that during navigation, where the view of the map (the world-centered reference frame, or WRF) does not correspond with the forward view of users (the ego-centered reference frame, or ERF), they are more likely to make errors [30]. This is caused by the task of aligning two different reference frames in a mental transformation to navigate successfully [31–35]. Aretz and Wickens [36] provided evidence that mental transformation happens in two mental rotation stages. The first one is to mentally rotate the map so that the top aligns with the forward direction. The second one is to mentally tilt the map 90°, so that the mental image of the map aligns with the forward view. In north-up maps, it is almost always the case that the two mental rotations must be done to align the WRF and the ERF. In comparison, a forward-up map, which aligns the map orientation with the user's direction of travel, reduces cognitive load [37,38] because it eliminates the first mental rotation to match the map to their physical orientation. As a solution to this problem, automatic map rotation in digital maps enhances the user's sense of direction. It is a feature where the map orientation adjusts so that the user's forward direction is always up. Aretz findings show that ERF tasks, which pedestrian navigation is all about, are performed more effectively with automatic map rotation [39]. In an experiment conducted by Wickens et al [40], it was discovered, that the error rate of a tracking task was substantially lower with rotating maps than fixed maps. Rodes and Gugerty [41] confirmed these findings with their research on forward-up or track-up maps. Participants performed consistently better on tasks involving cardinal directions and route following. Even though automatic map rotation enhances the performance of route-following tasks, it can hinder the formation of a cognitive map due to its inconsistent display of the map [39,41]. Thus the two different map orientations can serve two different use cases, namely active navigation with a forward-is-up map and map exploration and overview tasks with a north-is-up map.

## 2.3 Estimated time of arrival

Goodchild [42] emphasises the role of real-time data in volunteered geographic information, highlighting how it enables users to contribute timely and relevant data that enhances the accuracy and usability of navigation systems. This integration of real-time data enables dynamic updates that reflect current conditions, such as traffic congestion or street closures, which are crucial for effective route planning. Similarly, Zheng et al. [43] discuss the importance of real-time data in computing, illustrating how it can benefit navigation apps by incorporating live information from various sources, including GPS, sensors, and user reports. This real-time integration supports more efficient and responsive navigation, offering users optimised routes that consider current traffic conditions, thereby reducing travel time. In the field of HCI, real-time feedback is identified as a principle of user interface design. Djajadiningrat et al. [9] emphasise that real-time progress updates are essential for helping users understand their status and the next steps to take. This principle is crucial for adaptive navigation systems, enabling dynamic rerouting in response to changes in the user's environment, such as unexpected obstacles or shifts in pedestrian traffic. Don Norman's Seven Stages of Action [44] further underscores the importance of continuous information flow between the user and the system. Norman highlights that users need to continuously perceive, interpret, and evaluate their actions to achieve their goals. Real-time location updates are vital in this process as they provide users with the necessary information to adjust their actions based on current conditions. Thus, the accuracy of the ETA is crucial in pedestrian navigation, particularly for users planning, e.g., to catch public transportation. Precise ETA information enables users to make informed decisions about their routes and thus ensure timely arrivals [45]. It furthermore enables transportation efficiency, through reducing travel costs, saving energy, and overall contributing to lower air pollution [46]. Updated ETA features align with standard design principles, significantly enhancing the user experience by enabling efficient route planning, delay avoidance, and informed decision-making. Minimising error rates in real-time information is crucial for pedestrian navigation apps, as inaccuracies significantly impact user satisfaction and effectiveness. Gooze et al. [47] found that users expect high accuracy, with a margin of error of only 4–6 minutes. Therefore, precise real-time data is essential for user trust.

These findings suggest that integrating real-time progress updates and accurate ETA information is crucial for pedestrian navigation apps, enhancing usability, adaptability, and overall user satisfaction. This paper identified these and more relevant improvements to the RP navigation mode to make it a viable and effective alternative to the currently used TBT navigation mode for pedestrians.

Based on the introduced wayfinding strategies and the reviewed literature, a navigation app for pedestrians should integrate the following key features to enhance user experience: First, it should incorporate landmarks, route-based navigation, and survey knowledge to support various wayfinding strategies and improve navigation efficiency [14,16,23,24,48]. Second, it should feature automatic map rotation to align with the user's direction of travel, thereby reducing cognitive load [39]. Third, accurate ETA and real-time progress updates should be included to enhance user satisfaction and trust [45–47]. The RP mode, which provides an aerial (bird's-eye) view of the entire route, enables pedestrians to see key landmarks and their spatial relationships along the way. This visual context is essential for effective wayfinding as it enhances spatial knowledge and helps build a cognitive map [10]. Despite extensive research suggesting that the birds-eye view of the RP mode should be a viable alternative for pedestrian navigation, the focus of the industry clearly remains on the TBT mode.

## 3 Studies

In this paper, we therefore investigate people's stated preferences for, as well as their performance when adopting, the RP mode over the standard TBT navigation mode with three studies. To examine the relationship between users' preferences, navigation performance, and design opportunities, this research was structured into three complementary studies. Study 1 investigated users' stated preferences for Route Preview (RP) and Turn-by-Turn (TBT) navigation modes through a large-scale online survey conducted with a representative United States sample. This survey provided an initial understanding of users' attitudes, preferences, and self-reported behaviours when engaging with different navigation methods. Study 2 extended this investigation by assessing actual navigation performance and spatial learning outcomes in a controlled field experiment with student participants at Monash University. This study aimed to determine whether the RP mode, despite requiring greater user engagement, performs comparably to or better than the standard TBT mode in real-world conditions. Finally, Study 3 explored the design potential of the RP navigation mode through a participatory co-design workshop. In this study, users were actively involved in identifying and prioritising interface improvements that could enhance usability, comprehension, and engagement. Together, the three studies offer a comprehensive examination of how individuals perceive, perform with, and envision improvements to the RP navigation mode, providing valuable empirical and design-based insights for the development of future navigation systems.

### 3.1 Methods and ethics

Our research was reviewed and approved by the Monash University Human Research Ethics Committee (Project Title: Exploring Navigation Skills Using Different Methods, Project ID: 23126). The Committee confirmed that the research meets the requirements of the National Statement on Ethical Conduct in Human Research. All participants provided informed written consent prior to taking part in the study. We outline the specific methods used in subsections of the three studies.

### 3.2 Study 1: Understanding stated preferences

To understand people's preferences for either the RP or TBT navigation mode, we conducted an online survey with a representative U.S. sample of 298 participants through Prolific (https://www.prolific.com/) on June 27, 2024. The survey consisted of one open and fifteen closed-ended questions about people's navigation behaviour, their preferences, and demographic information.



While self-reported preferences may not fully align with actual behaviour, this initial survey provides an essential baseline for understanding users' perceived needs and attitudes toward different navigation modes. To reduce the gap between stated and actual preferences, the survey questions were designed around specific and concrete scenarios rather than abstract evaluations. This limitation is explicitly recognised and subsequently addressed through behavioural observation in Study 2.

**3.2.1 Method.** The survey was built in Unipark (https://www.unipark.com/en/) and is structured in four parts: navigation behaviour description, navigation mode preference, navigation frequency and demographics. Two attention checks were built into the survey to verify that participants gave valid answers. The full questionnaire can be found in Appendix.

The data collected through the survey were analysed using descriptive statistics and binomial tests to examine preferences in navigation modes across various scenarios. While the initial sample was representative of the U.S. population, some participants were excluded from the final analysis based on specific criteria to ensure the relevance of the data to pedestrian navigation. Participants who never used navigation apps (37), indicated that they didn't see a difference between the two navigation modes (10), never walked as a mode of transportation (40), or never used navigation apps while walking (48) were excluded from the final sample. These exclusions were necessary to focus the analysis on participants with relevant navigation experience and understanding. After applying these exclusion criteria, the final sample consisted of 222 participants (75% of the original sample). This final sample was used for all subsequent analyses.

**Demographics:** The survey was completed by 222 participants: 101 female, 115 male, 5 diverse, and 1 preferred not to state their gender. The mean age was 43.9 years, ranging from 18 to 74 years (*median* = 43.0, *SD* = 15.1). Regarding formal education, 49.6% had a bachelor's degree, followed by a high school diploma (29.7%) and a master's degree (16.7%). 3.6% had a doctoral degree, and only one person had no formal education (0.45%). With 55.9%, the leading operating system used by the participants is iOS by Apple. The use of an Android system was reported by 44.1% of the participants. Regarding the navigation app, 74.8% of the participants reported using Google Apps. Apple Maps is used by 20.7%, and "other" apps by 4.5% of the respondents. A majority of the participants (56.3%) self-reported their proficiency using navigation apps as advanced. Intermediate was the second most given answer (22.5%), while expert was the third most picked option (20.3%). Only 2 participants (0.9%) self-reported as beginners.

To increase ecological validity, participants were asked to recall their most recent navigation experience before answering preference questions, grounding their responses in actual use cases rather than hypothetical scenarios.

**Frequency of Transportation Modes:** Table 1 presents the responses of participants regarding how frequently they use different modes of transportation and how often they use navigation apps while moving. The "never" category for walking and use of navigation apps while walking shows 0% due to the data preparation that was mentioned before. The results show a clear contrast between the two transportation modes. While driving is a daily activity, walking isn't as common. Additionally, navigation apps are used less frequently while walking, with most individuals relying on them only occasionally. In contrast, driving sees a higher dependency on these apps, with a majority of respondents using them regularly.

**3.2.2 Results.** In this subsection, we present the results of the survey, which include the 222 participants remaining after the previous data preparation step.

**Mode Preference:** 43.7% (97 participants) of the respondents chose the RP as their preferred navigation mode, and 56.3% (125 participants) selected TBT. A binomial test showed no statistically significant ($p$ = 0.068) for the observed preference. For Apple Maps, the preference for TBT is higher than across the full sample, with 69.6%, while for Google Maps, it is slightly lower, with 53.0% preferring TBT navigation. To examine if the preferred navigation mode differs significantly between demographic groups and navigation scenarios, we conducted a series of Chi-Square and binomial tests. We tested for differences based on age groups, gender, user operating system, and type of navigation app (Apple Maps vs. Google Maps). None of these factors showed statistically significant differences in navigation mode preference.

**Mode Preference in Scenarios:** When navigating in familiar areas, a majority of participants (169 / 76.1%) preferred the RP mode, while a minority (53 / 23.9%) chose the TBT mode. A binomial test found a significant preference



**Table 1. Study 1: Frequency of transportation mode usage and navigation app use.**

| Question | Response | Participants |
|---|---|---|
| Mode of transportation: Walking | Daily | 32 (14.4%) |
| | Several times a week | 80 (36.0%) |
| | Once a week | 43 (19.4%) |
| | Once a month | 67 (30.2%) |
| | Never | 0 (0.0%) |
| Use of navigation apps while walking | Always | 8 (3.6%) |
| | Often | 35 (15.8%) |
| | Sometimes | 111 (50.0%) |
| | Rarely | 68 (30.6%) |
| | Never | 0 (0.0%) |
| Mode of transportation: Driving | Daily | 95 (42.8%) |
| | Several times a week | 87 (39.2%) |
| | Once a week | 24 (10.8%) |
| | Once a month | 7 (3.2%) |
| | Never | 9 (4.1%) |
| Use of navigation apps while driving | Always | 27 (12.2%) |
| | Often | 91 (41.0%) |
| | Sometimes | 80 (36.0%) |
| | Rarely | 15 (6.8%) |
| | Never | 9 (4.1%) |

for RP mode in familiar areas ($p < 0.001$). In unfamiliar areas, the preference shifted, with 156 participants (70.3%) preferring the TBT mode and only 66 people (29.7%) choosing the RP mode. A binomial test found a significant preference for TBT mode in unfamiliar areas ($p < 0.001$). These results suggest that the familiarity of the area has a strong influence on users' navigation mode preferences, with a clear tendency towards RP in familiar areas and TBT in unfamiliar areas.

For longer routes than usual (based on the participants' self-assessment), 116 participants (52.3%) preferred the TBT mode, whereas 105 participants (47.3%) selected the RP mode. This difference was not statistically significant (binomial test, $p = 0.491$). Similarly, when navigating shorter routes than usual (based on the participants' self-assessment), 117 respondents (52.7%) preferred the RP mode, compared to 104 participants (46.8%) who chose the TBT mode, again with no significant difference (binomial test, $p = 0.409$). For the total of 222 participants using pedestrian navigation apps, 14.4% prefer exclusively RP and 12.6% prefer exclusively TBT navigation. The remaining 73.0% of participants changed their preferred navigation mode based on different scenarios.

**Features for Route Preview:** Lastly, we asked participants to report features that would improve upon the current implementations of the RP mode. We performed a frequency analysis on the answers. Landmarks were by far the most desired features by the respondents, while the other four top features were mentioned only a few times.

1. Landmarks (mentioned 27 times)

2. Street View (mentioned 8 times)

3. Points of Interest (mentioned 7 times)

4. Customisable (mentioned 5 times)

5. Automatic Map rotation (mentioned 3 times)



**Summary:** The survey revealed that a considerable share of participants actually preferred the RP mode over the TBT mode. While this preference suggests potential areas for enhancing current RP navigation designs and justifies further exploration, our subsequent study first examined whether there are measurable differences in performance between the two navigation modes.

### 3.3 Study 2: Understanding actual navigation performance

To warrant substantial changes and improvements to the RP navigation method, it is not sufficient to just show that many people prefer this navigation mode, especially in familiar environments. It is also necessary to show that this navigation mode does not negatively affect navigation performance. In this second study, we investigated the differences in navigation performance between RP and TBT navigation modes with 195 participants in a field study.

**3.3.1 Method.** This section outlines the methodology employed in our study, detailing our study design, metrics, data preparation and further statistical foundation.

**Study Design & Procedure:** We recruited 218 students who had recently enrolled at Monash University in Melbourne, Australia, as our participants. All participants took part in the experiment within three weeks to ensure similar outdoor and infrastructural conditions between the 21st of March 2023 and the 11th of April 2023. We followed a citizen-science approach [49,50], where the participants would perform the experiment on one another: one being the navigator using the app, the other monitoring the navigator (i.e., the "experimenter"). To ensure the reliability of the experiment, we conducted a pilot to verify that the instructions were easy to follow. We also prepared video tutorials for each task and made both a briefing session and assistance available to anyone unsure about the requirements. This is in line with best practice for citizen science experiments [51].

Our participants were assigned to one of two conditions: TBT navigation or RP navigation. Participants were randomly assigned one of 20 predefined routes located near the campus. The 20 routes were selected using the Google Maps API, with the route length kept between 1400 and 1600 meters, featuring 9–11 manoeuvres—a common setup for outdoor navigation studies [13] to ensure consistency. A manoeuvre is defined as a navigation action to take for the current step (e.g., turn left, merge, straight, etc.). The experiment was conducted using the current version of Google Maps (version 11.69.0401 for Android and version 6.57.0 for iOS), which was used by all participants. A GPS recording of the route navigated was taken in the background, using OSMAnd. Whilst travelling the route, experimenters were instructed to walk within 10 meters of the participant and monitor them closely. They noted down any navigation error, where a participant accidentally left the route, as well as the number of times participants looked at their smartphones. Immediately after completing the journey, the navigator was asked to complete a route sketching task [52,53], where they would sketch the route as they remembered it. The participants also completed a demographic survey, whose results are summarised in Table 2.

We recognise that TBT navigation was not originally developed to support spatial learning, as it was designed primarily for in-car navigation. Previous research indicates that TBT requires relatively low cognitive engagement compared with RP, which involves more active processing of spatial information and may therefore facilitate stronger spatial learning outcomes. Nevertheless, our comparison reflects realistic navigation conditions, in which users are often required to recall routes regardless of the navigation mode employed. To ensure a comprehensive and balanced evaluation, several methodological controls were implemented.

First, we selected performance metrics capturing both navigation efficiency, including Navigation Errors and Phone Glances, and spatial learning, assessed through the Route Sketching Task using the metrics Correctly Indicated Turns and Correctly Indicated Directional Changes. Second, we recruited participants with minimal familiarity with the study area, focusing on students who had recently enrolled at Monash University. Third, a pre-study familiarity assessment (summarised in Table 2) indicated that 44.11% of participants rated themselves as unfamiliar or very unfamiliar with the routes and over 69.2% including the neutral participants. Finally, we controlled for familiarity effects through multiple linear regression analysis (Table 3), which confirmed that increased familiarity was significantly and negatively associated with the number of phone glances, providing a validity check for our data. Although the Route Sketching Task may inherently favour RP due to its interface encouraging higher cognitive engagement and spatial encoding, it also reflects a realistic

**Table 2. Study 2: Participant demographics, preferences, and age statistics.**

| Metric | Description/Response | Participants |
|---|---|---|
| Age | Mean: 19.38 years, Median: 19, Std Dev: 1.87 | 195 |
| Gender | Male | 138 (70.77%) |
| | Female | 57 (29.23%) |
| Navigation app usage | More than once per week | 75 (38.46%) |
| | Once per week | 49 (25.13%) |
| | Every Day | 40 (20.51%) |
| | Once per month | 28 (14.36%) |
| | Never | 3 (1.54%) |
| Familiarity with route | Unfamiliar | 73 (37.44%) |
| | Neither familiar nor unfamiliar | 49 (25.13%) |
| | Familiar | 46 (23.59%) |
| | Very unfamiliar | 13 (6.67%) |
| | Very familiar | 14 (7.18%) |
| Navigation preference | TBT navigation | 102 (52.31%) |
| | Route Preview | 93 (47.69%) |

**Table 3. Study 2: OLS regression results. The regression analysis tested the significant influence of confounding variables in our study. This analysis included gender, age, familiarity with the experiment environment, the frequency of how regularly participants use mobile navigation apps, and their personal preference for TBT or RP navigation. Statistically significant results are marked with asterisks: $*p < 0.05$, $**p < 0.01$..**

| Variable | Navigation Condition | Gender | Age | Familiarity | Frequency | Navigation Preference |
|---|---|---|---|---|---|---|
| Navigation Errors | 0.2395 (0.308) | −0.1441 (0.483) | −0.0987 (0.051) | −0.0399 (0.651) | −0.0035 (0.970) | 0.0180 (0.939) |
| Phone Glances | 1.9508 (0.313) | −2.2200 (0.190) | −0.5127 (0.217) | −2.2918 (0.002)** | −0.2860 (0.712) | −3.2453 (0.094) |
| Turns | −1.2160 (0.727) | −2.2429 (0.463) | 0.4857 (0.516) | −0.1223 (0.926) | 2.0508 (0.144) | −0.9695 (0.781) |
| Directional Changes | −3.0843 (0.329) | 5.3317 (0.055) | 0.3966 (0.558) | 0.0453 (0.970) | 3.0503 (0.017)* | −1.7433 (0.581) |

situation in which users must navigate without digital assistance or communicate routes to others. This measure is therefore critical for evaluating spatial knowledge acquisition, which represents a central advantage of the RP mode.

**Metrics:** Our primary goal was to understand how navigation performance, attention during navigation, and spatial learning differ between TBT navigation and RP navigation. Given mixed findings in the literature in terms of the performance of those two modes, we did not setup specific hypotheses. However, we define four key metrics that will serve as performance indicators common for navigation experiments [12,52,54–56]:

- **Navigation Errors**: This metric quantifies the number of mistakes made by the participant during the navigation task. An error is defined as any deviation from the predefined route. A lower number of errors indicates higher navigation performance.

- **Phone Glances**: This metric measures the frequency with which participants looked at their smartphones during the navigation task. Frequent glances may indicate greater reliance on the navigation system and potentially reduced awareness of the surrounding environment.

- **Correctly Indicated Turns**: This metric gauges the accuracy of the participants' route sketching task by counting the number of turns (left, right, etc.) correctly indicated on their sketched maps. A higher number suggests better spatial learning and memory of the route.

- **Correctly Indicated Directional Changes**: Similar to "Correctly Indicated Turns," this metric also evaluates the accuracy of the participants' route sketching task. However, it focuses on capturing broader directional changes, such as merging or going straight, as opposed to just turns. Like the previous metric, a higher count suggests better spatial understanding and memory of the route.

These metrics collectively provide insights into the navigation performance, attentional engagement, and spatial learning outcomes of participants, thereby enabling a comparison of the effectiveness of TBT navigation and RP navigation.

**Data Preparation:** Before our evaluation, we used a two-step approach to clean and prepare our data. In the first step, we removed entries identified as faulty data due to participants not having thoroughly recorded or submitted their data. For this, we used indicators such as map sketch results of 1000% or over 100 navigation errors. For each such value that we noticed, we reviewed any comments made by the experimenter that suggested there had been an execution error with the experiment. Only if these two indicators matched were we able to remove the participants from the dataset. Furthermore, we removed entries that stated they had to use a custom route instead of the provided ones, due to participating remotely, which resulted in routes with different lengths and numbers of manoeuvres outside the defined range for our study. After cleaning the data, 208 out of 218 participants remained in the dataset. In a second step, we identified outliers and removed those with a Z-score above 3 to exclude extreme outliers for statistical significance testing. After removing these outliers, 195 participants remained in the final dataset. All statistical tests in our analysis were conducted on both the raw and cleaned datasets, yielding consistent results.

**Statistical Methods:** To test our data for normality, we applied a Shapiro test to our data for each condition and metric, which resulted in a $p-value$ < 0.01 for all results. Thus, we assume that our data is not normally distributed. Since our data are also unpaired, the Mann–Whitney U test was used to compare results between the two conditions, unless otherwise specified. Effect sizes are calculated using the rank-biserial correlation (denoted with $r$ in the results). Furthermore, we used multiple linear regression to understand the relationship between the predictor variable (the experiment condition), our dependent variables, and other covariates.

**Power Analysis:** We used the G*Power software to conduct a power analysis for our study. According to Robertson and Kaptein [57], effect sizes for non-parametric tests like the Mann-Whitney U test typically range from 0.1 for a small effect, 0.3 for a medium effect, and 0.5 for a large effect. Given our study design and common assumption for power and significance, detecting a large effect would require 134 participants, a medium effect would require 368, and a small effect would require 3290.

With 195 participants, our study is sufficiently powered to detect a large effect. We chose to focus on detecting a large effect because TBT navigation is the default and far more prevalent mode in mobile navigation apps compared to RP. Thus, only a large difference in effectiveness between these modes would warrant the current design choice of TBT being the default. A small or medium effect, while potentially statistically significant, would not be sufficient to advocate for such a decision; it would rather argue for the co-existence of both modes to cater to user preferences.

**Balanced Random Assignment to Conditions:** Our study utilised random assignment to distribute participants across two distinct navigation conditions: RP and TBT. To validate the effectiveness of this randomisation process, we examined key demographic and behavioural variables. The mean age in the RP condition was approximately 19.24 years, while it was 19.45 years in the TBT condition ($T = -0.90, p-value = 0.37$). Both conditions displayed a balanced gender distribution between each other with around 30% females and 70% males. 78% of participants in the RP condition and 81% in the TBT condition preferred the navigation method they were assigned to. Although this could be influenced by the timing of the post-experiment demographic questionnaire, it is noteworthy that about 20% of participants in each condition preferred the opposite navigation method. This suggests that any potential bias introduced is likely balanced across both conditions. Overall, these findings affirm the success of the random assignment to conditions.

**3.3.2 Results.** After cleaning the data and removing extreme outliers, a total of 195 participants (out of the initial 218) contributed valid data to the study. Their demographic information and participant preferences are summarised in Table 2. In the following section, we present our results of the study.

**Navigation Errors:** During the navigation task, the experimenter noted any navigation errors made by the participant. On average, participants made more errors in the TBT condition (*mean* = 1.27, *median* = 1, *SD* = 1.41) compared to the RP condition (*mean* = 1.05, *median* = 1, *SD* = 1.13). Even though this was not statistically significant ($U$ = 5010, *p–value* = 0.49, $r$ = –0.06), there is a trend towards TBT navigation resulting in more errors (see Fig 1). In a OLS regression model, none of the variables were found to be statistically significant control variables for navigation errors. For detailed coefficients and *p*-values, please refer to Table 3.

**Attention During Navigation:** The experimenter noted down the number of times the participant looked at their phone. On average, participants looked at their phone more often in the RP condition (*mean* = 13.08, *median* = 11, *SD* = 11.61) than in the TBT condition (*mean* = 13.01, *median* = 11, *SD* = 10.25). While this was not statistically significant ($U$ = 4528, *p–value* = 0.58, $r$ = 0.05), we see an indication that RP navigation results in more attention on the mobile navigation device (see Fig 1). This observation, however, fits the notion of related work that RP navigation might demand more elaborate processing of spatial information [10] and thus lead to more increased phone glances. An OLS regression model for the phone glances (looks at the navigation device) found that the variable 'Familiarity' was statistically significant ($p < 0.01$) and negatively correlated with the number of glances at the phone. This suggests that participants who are more familiar with the area tend to look at the navigation device less frequently, which is a logical finding and thus serves as a sanity check for our data. Table 3 shows a complete list of coefficients and *p*-values. A subsequent correlation analysis revealed a statistically significant negative correlation between 'Familiarity' and the number of 'Phone Glances' at the navigation device ($r$ = –0.234, $p < 0.001$). This suggests that as participants' familiarity with the area increased, they tended to look at the navigation device less frequently.

**Route Sketching Task:** After completing the navigation task, participants were asked to sketch the route they had just walked on a piece of paper. The evaluation of the sketch consisted of two parts: the percentage of correctly indicated turns and the percentage of correctly indicated directional changes.

**Correctly Indicated Turns:** Participants, on average, correctly indicated more turns in the RP condition (*mean* = 87.14%, *median* = 87.5, *SD* = 19.57) than in the TBT condition (*mean* = 86.09%, *median* = 87.5, *SD* = 18.48). There was no statistically significant difference between the two ($U$ = 4632, *p–value* = 0.77, $r$ = 0.02), but the trend points

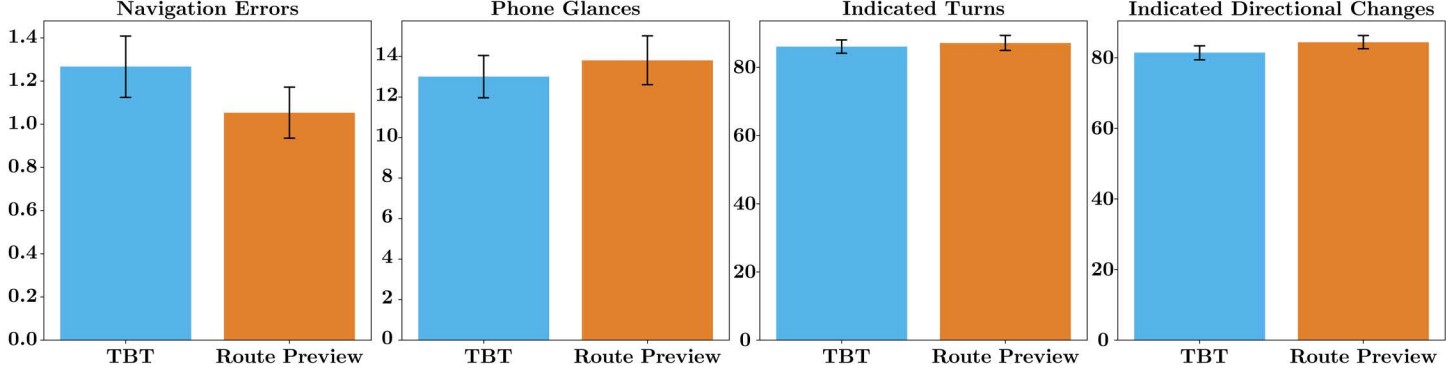

**Fig 1. Study 2: Mean values of key navigation metrics between the two conditions: turn-by-turn navigation and route preview navigation.**

towards RP resulting in better route sketching abilities. The regression model for indicated turns found no variables to be significant control variables for the correct number of turns indicated. For further details, please see Table 3.

**Correctly Indicated Directional Changes:** For the correctly indicated directional changes, participants also correctly indicated more turns in the RP condition (*mean* = 84.40%, *median* = 87.5, *SD* = 16.73) than in the TBT condition (*mean* = 81.46%, *median* = 85.7, *SD* = 18.28).

An OLS regression model found that the variable 'Frequency of Using Navigation Apps' was statistically significant ($p < 0.05$) and positively correlated with the percentage of correctly indicated directional changes. This suggests that participants who frequently use navigation apps tend to make more accurate directional changes. For additional coefficients and *p*-values, refer to Table 3. A subsequent correlation analysis revealed a statistically significant positive correlation between 'How often users use mobile navigation apps' and the percentage of 'Correctly Indicated Directional Changes' ($r = 0.159$, $p = 0.02$). This suggests that as participants' amount of general mobile app usage increases, their ability to correctly indicate turns in our route sketching task improves.

**Analysis of Navigation Method Preference:** We investigated the potential confounding factor of whether participants used or did not use their preferred navigation method. To assess the impact of this factor, we conducted Mann-Whitney U tests for our four dependent variables: number of errors made, number of times the participant looked at the navigation device, the percentage of correctly indicated turns, and the percentage of correct directional changes. Our analyses revealed no statistically significant differences between the group of participants who used their preferred navigation method and those who did not, across all four variables. This suggests that the preference for a particular navigation method did not significantly affect the outcomes in our study.

**Post-hoc Power Analysis:** Our analysis across all dependent variables revealed small effect sizes, which have critical implications for the interpretation of our results. For instance, in the case of navigation errors, a post-hoc power analysis demonstrated that with an effect size of 0.06 and our current sample size, the achieved power was only 0.07. To detect an effect of this magnitude as significant, a substantially larger sample size would be required.

Interestingly, the TBT navigation condition actually performed worse than the RP condition, a finding that is surprising based on current assumptions in the field. However, it is crucial to note that the observed effect sizes are not only statistically insignificant but also practically negligible. Even if statistical significance could be achieved with a larger sample, the practical impact of these effect sizes would be minimal. This reinforces the idea that the prevailing implementation of TBT navigation may need to be re-evaluated, especially given its counterintuitive, poorer performance in our study.

### 3.4 Study 3: Understanding the potential of route preview

Studies 1 and 2 demonstrated that, first, a substantial proportion of users prefer RP navigation, particularly in familiar contexts, and second, RP performs comparably to TBT despite the absence of several key features. These findings prompt a central design question: could RP navigation outperform TBT if supported by appropriate enhancements? Study 3 addresses this question through a co-design process that engages users in identifying and prioritising potential improvements. Rather than implementing researcher-defined modifications, this participatory approach ensures both ecological validity and user-centred design. The study yields directly applicable insights for practitioners seeking to implement or extend the RP navigation mode.

**3.4.1 Method.** We used the design studio method to conduct our workshop [58]. The goal of the design studio was to develop concrete interface designs for features that participants would like to see in combination with the RP navigation mode. The workshop was organised into several phases. It began with an introduction by the facilitators, who were two of the paper's authors. This introduction included an overview of the workshop schedule and a brief presentation of the research, summarising the related work discussed in Section 2. During the interactive parts, participants generated ideas, developed them further, reflected on their relevance, and finally prioritised them. This process resulted in the identification of the most crucial features for enhancing the user interface. The workshop, which lasted three hours, took place on June



12, 2024. We recruited five participants [59,60], all of whom signed a consent form. The workshop was audio- and video-recorded.

**Demographics:** Participants were, on average, 26.4 years old ($min = 23, max = 28$). The group consisted of two female and three male participants. Their educational backgrounds ranged from high school diplomas to doctoral degrees, offering a diverse range of perspectives. Three participants used Apple's iOS, while two used an Android system.

**3.4.2 Results.** This section presents the exploratory findings of the workshop. The following subsections outline the design implications for Landmarks, ETA Updates, and Map Orientation, beginning with the highest-priority topic identified by the workshop participants.

**Landmarks:** Landmarks emerged as a crucial feature in our design studio workshop, with participants unanimously endorsing their integration into the RP mode. Defined as important spatial features used as reference points [61], landmarks play a vital role in everyday navigation strategies. Participants recounted using landmarks to orient themselves in familiar cities and to correct navigational errors:

> "When I'm looking for a restaurant in the city that I haven't been to before but know the city relatively well, I approach it differently. In such a case, I would simply open Google Maps, enter the location, switch to satellite view, and then look for familiar landmarks nearby" (P2).

> "Sometimes it shows the wrong direction, but then I look at what buildings are on the map, compare them a bit, and then I get a sense of direction and follow the route" (P1).

All participants incorporated landmarks into their design sketches, albeit with varying approaches. Some suggested displaying all landmarks along the route, while others proposed showing only the next landmark to minimise distraction. A novel "Bubble" concept was introduced, where interactive 360° street view pop-ups would appear at each turn (see Fig 2). This idea was well-received and further developed:

> "I also find it important to show Bubbles at critical locations. For me, I would also like to be able to easily drag and drop Bubbles onto the user interface" (P1).

P5 added that it would be helpful to drag the Bubble across the map to get the 360° Street view everywhere the user wants. However, P4 raised a concern:

> "[A user] might miss a landmark that they wanted to see, and it restricts the navigator unnecessarily" (P4).

In the final prioritisation, the "Bubble" feature emerged as the most favoured. In total, landmark-related features received the overall highest prioritisation, underscoring their perceived importance in enhancing the RP navigation mode.

**ETA Updates:** Throughout the workshop, the idea of estimated time of arrivals evolved into a nuanced discussion about intention-based navigation. While participants didn't explicitly mention ETA updates in their daily navigation habits, the feature sparked creative ideas during the sketching phase. P5 introduced a novel concept that went beyond simple ETA updates: P5 introduced a feature where users get different routes and ETAs based on 4 different walking speed options (see Fig 2). A user who chooses slow walking for example gets a route that guides them through the most scenic way, while a user who chooses speedwalking will get the fastest route. This idea was well-received, with P2 commenting:

> "Usually, there is a direct route and one that might be a bit slower, similar to what you know from driving a car, but I find these walking speeds very interesting."

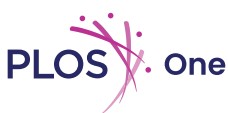

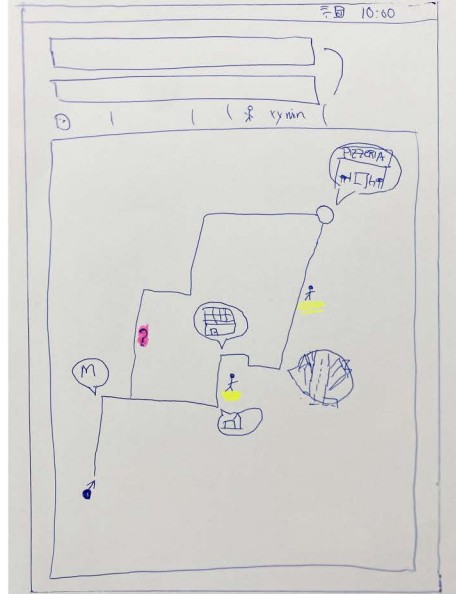 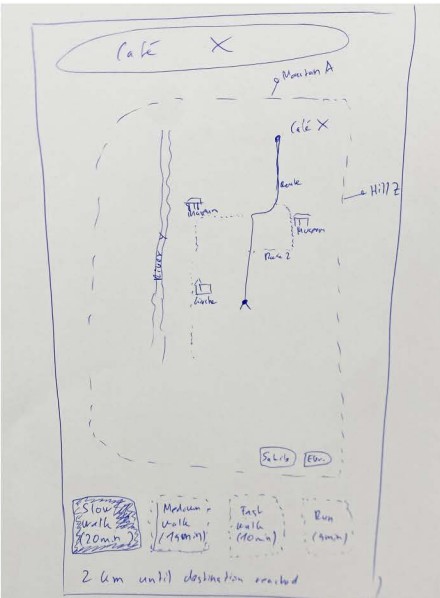 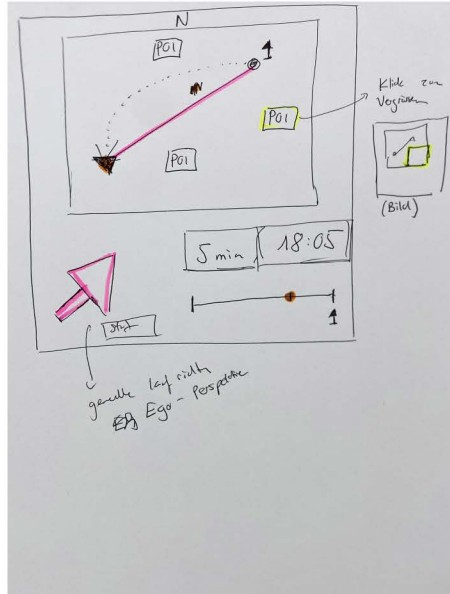

**Fig 2. Study 3: Sketches from the workshop.** (left) landmarks at important turns called "bubbles", (middle) choosing different walking speeds, (right) a small arrow to indicate the direction towards the destination.

P2 also suggested that alternative routes for different speed settings should be visible, allowing users to make informed decisions based on the path. Furthermore, they commented on how the map could be designed based on the users intention of discovering the environment or wanting to efficiently reach a destination:

"I have also seen landmarks primarily as orientation aids, in the sense that I only need them at a turn and not otherwise. However, there may also be the possibility of having this as an option. For example, they could be displayed by default when I need to turn, but also that I could filter by interests. If I enable discovery mode, additional landmarks may appear based on my interests. You should be able to switch between an orientation mode and a discovery mode."

This statement shifted the focus from speed-based navigation to intention-based navigation. It combined elements of P5's speed option idea with a new emphasis on user intentions, differentiating between goal-oriented navigation and exploratory wandering. Further refinement led to the "Discovery or Speed" feature, where users choose between the fastest route (Speed mode) or a route showcasing interesting landmarks (Discovery mode). In Discovery mode, a filter would match personal interests, displaying only relevant landmarks. Speed mode, conversely, would show only navigation-critical landmarks. While some ideas, such as using a voice assistant for ETA updates or implementing a progress bar, were proposed and subsequently discarded, the "Discovery or Speed" feature gained significant traction. It transformed the initial concept of ETA updates into a comprehensive navigation approach that adapts to user intentions, with ETA serving either as the primary focus or as background information depending on the chosen mode.

**Map Orientation:** The discussion on map orientation revealed a tension between cognitive load and spatial awareness. During the reflection on personal navigation habits, a clear preference emerged. Four out of five participants (P1, P2, P4 & P5) described using RP with a north-up map orientation. P5's description exemplifies this approach:

"I open Google Maps, enter the café or place I want to go to, and search. Then I check if it is the right café that I want, and if it is, I click on the route. It usually suggests several routes, and then I see if it's correct, and then I head in the

direction of the route. I don't click on the start button. Instead, I occasionally take out my phone, check if I walk in the right direction or see if I've gotten lost."

Only P3 reported using the turn-by-turn (TBT) mode with automatic map rotation, citing difficulties with spatial orientation:

"I don't really find my way around places, even those I know well. So, for me, it's the same that whenever I walk from the university to the train station, I always have navigation turned on".

The sketching phase revealed diverse approaches to map orientation. While P2, P3, P4, and P5 opted for a north-up map in bird's-eye view, P1 proposed a 3D mode as the main navigation tool, with the option to switch to a north-up bird's-eye view. From this concept emerged a mini-map feature. P1 described a user interface with two mini-maps at the bottom: one showing the entire route in bird's-eye view with the user's position indicated, and another displaying a fraction of the route. This idea was well-received, with P2 declaring it "very intuitive and thus a good addition." P2 proposed a hybrid approach, suggesting the addition of a TBT arrow to the RP mode (see Fig 2):

"Right now, it's a map oriented to the north, so I still have to adjust a bit. Even though I don't really like the TBT mode, I would still like to have an arrow that shows me [...] when to turn right".

This concept was further developed by P1 and P5, who suggested displaying the arrow on the phone's lock screen and having an option for the arrow to always point towards the final destination. During the refinement phase, the group reached a consensus that a mode switch between north-up and 3D views might be beneficial, though the default setting remained undecided. The mini-map concept was streamlined to a single map, which could also serve as a progress indicator. In the final prioritisation, participants showed a clear preference for the mini-map concept and the arrow/compass idea. Interestingly, an initially proposed AR feature, despite initial excitement, was discarded due to a lack of interest. These results, while based on a smaller sample size, provide valuable exploratory findings suggesting a user preference for subtle, informative aids like mini-maps and directional indicators over more immersive but potentially overwhelming features like AR.

## 4 Discussion

Route preview (RP) navigation is just as commonly used compared to the more refined TBT navigation mode in pedestrian navigation applications. The results of our online survey supported that in certain scenarios or contexts, like being familiar with the environment, RP navigation is even more likely to be used. In study 2 involving 195 participants, we surprisingly found that the Turn-By-Turn (TBT) navigation and RP modes did not significantly differ in performance measures like navigation errors, the number of times participants looked at their phone, or their ability to correctly sketch turns and directional changes in a map. Even so, the RP mode offers significant less navigation features that are proven to improve the navigation performance, on the margin, it performed even better than TBT.

This also implies that the RP feature, often considered secondary to TBT navigation, has the potential to serve as a viable alternative for navigation, supporting a richer cognitive engagement and spatial understanding, as discussed in previous literature [1–3]. This realisation necessitates the consideration of specific design implications to optimise the RP feature. We explored such design implications in a design studio workshop with five participants and discussed them further in the following sections.

### 4.1 The role of route-preview

The route-preview used in modern mobile navigation applications is a well-versed tool for visualising and exploring possible routes from one location to another and therefore facilitates a widely used feature, namely getting directions [4]. The results of our studies show that even though it is not its main use case, the route-preview is just as often preferred as an active



navigation aid as the well-known TBT navigation method. One of the unique advantages of RP is that it naturally encourages greater cognitive engagement [10]. Unlike TBT navigation, where the user is largely a passive follower of instructions [62], RP requires users to actively interpret the map and make decisions. This active role could potentially foster a better spatial understanding and situational awareness [10,11], although our study did not show a significant improvement in this regard. Route preview empowers users with a greater sense of autonomy and control over their navigation experience. Users can explore alternative routes, preview upcoming turns, and mentally prepare for the route. This is particularly useful for those who like to have a broader understanding of their environment rather than just getting from A to B. Ultimately, RP is adaptable to various tasks and user preferences. For instance, users who prefer not to be interrupted by frequent instructions may find RP more accommodating. Similarly, those involved in activities where auditory instructions could be disruptive, like walking in a group, might find RP to be a more suitable and socially accepted navigation aid.

### 4.2 Implications for design

Given its unique benefits and challenges, there is a compelling case for designing RP features that are as robust and user-friendly as TBT options. Our findings suggest that RP, if designed with specific enhancements such as displaying relevant information about nearby landmarks, adapting the travel mode to the user's intentions, and adding information for better orientation, could improve the user experience and possibly even outperform TBT navigation. By recognising and leveraging the distinct roles and benefits of RP, designers and developers have an opportunity to create more versatile, adaptable, and user-friendly navigation systems for pedestrians. The results of our survey showed that the preference for RP changes based on familiarity with the route. At the same time, participants in the design studio workshop highlighted the need for different navigation modes like discovery and speed modes. Such insights strongly point to the need for adaptive interfaces based on individual user needs and context. Thus, we argue that not only should the RP serve as a viable alternative to TBT navigation, but its functionality should also be relying on the context in which it is used.

This research started out comparing two different navigation modes: RP and TBT. After collecting results from a total of 422 participants, it becomes clear that the solution will not be a simple alternative to the status quo, but an adaptive solution that incorporates the best of both worlds. Depending on the needs of pedestrians, whether they are in a hurry, walking familiar routes, or exploring new places, a mobile navigation application should adapt to these different goals. This involves combining the different features currently existing in TBT navigation and RP navigation, while also extending them with concepts like landmarks to further increase user satisfaction and usefulness.

This paper contributes a good foundation for arguing that users want change in this domain, and we deliver first design implications to guide the direction so that this change can find its way into the everyday products we already use. Fig 3 illustrates a first interpretation of our findings. It is now upon future work to refine these concepts further and build the next generation of mobile navigation technologies for pedestrians, as they have been mimicking the user experience of car drivers for too long.

Several considerations regarding our research design merit attention. First, Study 1 relied on self-reported preferences, which may not fully reflect actual usage patterns; this was addressed through behavioural observation in Study 2. Second, the performance metrics in Study 2 were designed to capture spatial learning, which aligns with the growing importance of independent wayfinding in pedestrian navigation where continuous device access is not always possible. Third, the sample in Study 2 primarily comprised university students, which should be considered when interpreting generalisability. Finally, the design implications from Study 3, while directly informed by user input, would benefit from further empirical evaluation through implementation and testing.

### 5 Conclusion

This research interrogates the assumption that turn-by-turn (TBT) navigation, originally designed for vehicular contexts, constitutes the optimal solution for pedestrian wayfinding. Through a systematic investigation of user preferences, performance metrics, and design opportunities, we demonstrate that route-preview (RP)—despite often being considered a

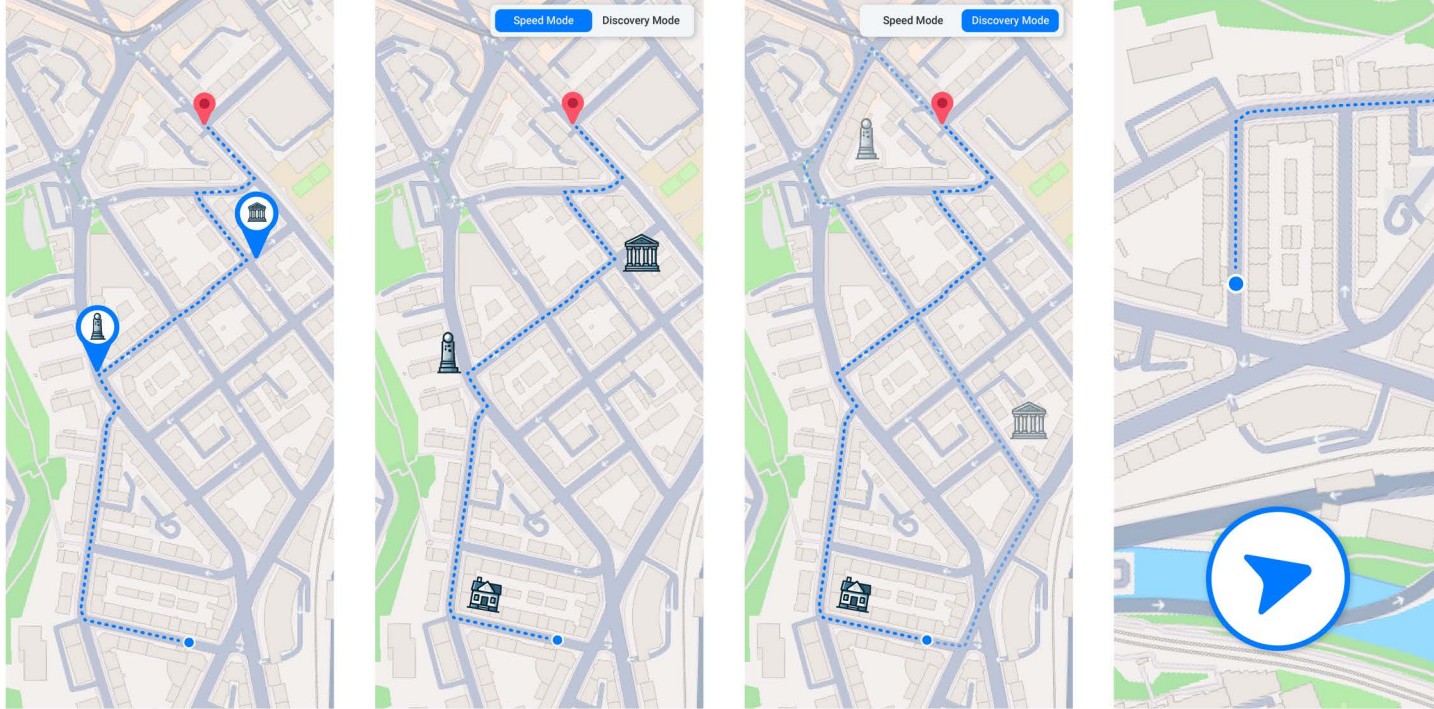

**Fig 3. Study 3: Illustrations made based on the sketches and insights from the workshop.** (left) landmarks at important turns called "bubbles", (middle left) speed mode getting important landmark information at relevant turns, (middle right) discovery mode giving optional detours to visit interesting landmarks (right) a small arrow to indicate the direction towards the destination. The base map under the sketches was done with Mapbox. Map tiles and data remains Mapbox https://www.mapbox.com/about/maps) OpenStreetMap http://www.openstreetmap.org/about/ contributors, styled using Mapbox Studio.

secondary feature—yields comparable performance while fostering greater cognitive engagement and user autonomy. These findings suggest that the prevailing dominance of TBT in pedestrian navigation applications may reflect historical path dependence rather than demonstrable empirical superiority.

The development of navigation technologies has produced a diverse range of tools aimed at enhancing pedestrian mobility and spatial awareness. While TBT navigation has become the default option for pedestrians, largely through the transposition of vehicular design paradigms, this study reveals the largely untapped potential of RP features in mobile navigation applications. Our data indicate no significant differences in key performance metrics, including navigation errors, frequency of phone glances, and the ability to recall and sketch turns and directional changes, between TBT and RP modes. Notably, RP demonstrated marginally superior outcomes on several measures. These results challenge the conventional assumption that TBT navigation is inherently more effective for pedestrians, opening new avenues for designing navigation aids better aligned with pedestrian needs.

Beyond validating the functional efficacy of RP, our study highlights its capacity to support active cognitive engagement. Users who seek a more participatory role in navigation or enhanced autonomy may find RP particularly suitable. Furthermore, our findings support the integration of features such as contextually relevant landmark information, adaptive travel modes responsive to user intentions, and additional orientation cues to enhance wayfinding effectiveness.

In sum, this research encourages a re-evaluation of the design and functionality of pedestrian navigation systems. By emphasising the overlooked capabilities of RP, we provide a framework for developing more versatile, user-centric, and cognitively engaging navigation aids that move beyond the constraints of conventional TBT paradigms.



## Supporting information

**S1 File.** *S1 Data Study 1* **Contains the Data from Study 1.**
(XLSX)

**S2 File.** *S2 Data Study 2* **Contains the Data from Study 2.**
(CSV)

## Author contributions

**Conceptualization:** Gian-Luca Savino, Emanuel de Bellis, Reuben Kirkham, Johannes Schöning.

**Data curation:** Gian-Luca Savino, Emanuel de Bellis, Reuben Kirkham.

**Formal analysis:** Gian-Luca Savino, Emanuel de Bellis, Reuben Kirkham, Johannes Schöning.

**Funding acquisition:** Johannes Schöning.

**Investigation:** Reuben Kirkham, Johannes Schöning.

**Methodology:** Gian-Luca Savino, Emanuel de Bellis, Reuben Kirkham, Johannes Schöning.

**Project administration:** Johannes Schöning.

**Software:** Gian-Luca Savino, Reuben Kirkham.

**Supervision:** Johannes Schöning.

**Validation:** Gian-Luca Savino, Emanuel de Bellis, Johannes Schöning.

**Visualization:** Gian-Luca Savino.

**Writing – original draft:** Gian-Luca Savino, Emanuel de Bellis, Reuben Kirkham, Johannes Schöning.

**Writing – review & editing:** Gian-Luca Savino, Emanuel de Bellis, Reuben Kirkham, Johannes Schöning.

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
