## [Decision Letter · Decision Letter 0]

5 Sep 2025

Dear Dr. Schöning,

Thank you for submitting your manuscript to PLOS ONE. After careful consideration, we feel that it has merit but does not fully meet PLOS ONE’s publication criteria as it currently stands. Therefore, we invite you to submit a revised version of the manuscript that addresses the points raised during the review process.

We look forward to receiving your revised manuscript.

Kind regards,

Sukhjit Singh Sehra

Academic Editor

PLOS ONE

Journal Requirements:

2. Please include a complete copy of PLOS’ questionnaire on inclusivity in global research in your revised manuscript. Our policy for research in this area aims to improve transparency in the reporting of research performed outside of researchers’ own country or community. The policy applies to researchers who have travelled to a different country to conduct research, research with Indigenous populations or their lands, and research on cultural artefacts. The questionnaire can also be requested at the journal’s discretion for any other submissions, even if these conditions are not met.

Please find more information on the policy and a link to download a blank copy of the questionnaire here: https://journals.plos.org/plosone/s/best-practices-in-research-reporting.

Please upload a completed version of your questionnaire as Supporting Information when you resubmit your manuscript.

“GS and JS - Swiss National Science Foundation (SNSF) grant number 200021 207430.”

6. We note that Figures 1 and 4 in your submission contain map images which may be copyrighted. All PLOS content is published under the Creative Commons Attribution License (CC BY 4.0), which means that the manuscript, images, and Supporting Information files will be freely available online, and any third party is permitted to access, download, copy, distribute, and use these materials in any way, even commercially, with proper attribution. For these reasons, we cannot publish previously copyrighted maps or satellite images created using proprietary data, such as Google software (Google Maps, Street View, and Earth). For more information, see our copyright guidelines: http://journals.plos.org/plosone/s/licenses-and-copyright.

1) You may seek permission from the original copyright holder of Figures 1 and 4 to publish the content specifically under the CC BY 4.0 license.

2) If you are unable to obtain permission from the original copyright holder to publish these figures under the CC BY 4.0 license or if the copyright holder’s requirements are incompatible with the CC BY 4.0 license, please either i) remove the figure or ii) supply a replacement figure that complies with the CC BY 4.0 license. Please check copyright information on all replacement figures and update the figure caption with source information. If applicable, please specify in the figure caption text when a figure is similar but not identical to the original image and is therefore for illustrative purposes only.

7. Please ensure that you refer to Figure 3 in your text as, if accepted, production will need this reference to link the reader to the figure.

9. We note that there is identifying data in the Supporting Information file <data.zip>. Due to the inclusion of these potentially identifying data, we have removed this file from your file inventory. Prior to sharing human research participant data, authors should consult with an ethics committee to ensure data are shared in accordance with participant consent and all applicable local laws.

-Location data

Please remove or anonymize all personal information, ensure that the data shared are in accordance with participant consent, and re-upload a fully anonymized data set. Please note that spreadsheet columns with personal information must be removed and not hidden as all hidden columns will appear in the published file.

10. We note that this data set consists of interview transcripts. Can you please confirm that all participants gave consent for interview transcript to be published?

If they DID provide consent for these transcripts to be published, please also confirm that the transcripts do not contain any potentially identifying information (or let us know if the participants consented to having their personal details published and made publicly available). We consider the following details to be identifying information:

- Names, nicknames, and initials

- Age more specific than round numbers

- GPS coordinates, physical addresses, IP addresses, email addresses

- Information in small sample sizes (e.g. 40 students from X class in X year at X university)

- Specific dates (e.g. visit dates, interview dates)

- ID numbers

Or, if the participants DID NOT provide consent for these transcripts to be published:

- Provide a de-identified version of the data or excerpts of interview responses

- Provide information regarding how these transcripts can be accessed by researchers who meet the criteria for access to confidential data, including:

a) the grounds for restriction

b) the name of the ethics committee, Institutional Review Board, or third-party organization that is imposing sharing restrictions on the data

c) a non-author, institutional point of contact that is able to field data access queries, in the interest of maintaining long-term data accessibility.

d) Any relevant data set names, URLs, DOIs, etc. that an independent researcher would need in order to request your minimal data set.

For further information on sharing data that contains sensitive participant information, please see: https://journals.plos.org/plosone/s/data-availability#loc-human-research-participant-data-and-other-sensitive-data

If there are ethical, legal, or third-party restrictions upon your dataset, you must provide all of the following details (https://journals.plos.org/plosone/s/data-availability#loc-acceptable-data-access-restrictions):

1. A complete description of the dataset

2. The nature of the restrictions upon the data (ethical, legal, or owned by a third party) and the reasoning behind them

3. The full name of the body imposing the restrictions upon your dataset (ethics committee, institution, data access committee, etc)

4. If the data are owned by a third party, confirmation of whether the authors received any special privileges in accessing the data that other researchers would not have

5. Direct, non-author contact information (preferably email) for the body imposing the restrictions upon the data, to which data access requests can be sent

**Additional Editor Comments:**

I thank authors for submitting the paper. But unfortunately, the paper is not yet ready. I encourage authors to make substantial improvements to the manuscript and address all concerns of the reviewers.

Reviewers' comments:

Reviewer's Responses to Questions

**Comments to the Author**

1. Is the manuscript technically sound, and do the data support the conclusions?

Reviewer #1: Yes

Reviewer #2: No

Reviewer #3: Yes

2. Has the statistical analysis been performed appropriately and rigorously?

Reviewer #1: Yes

Reviewer #2: No

Reviewer #3: Yes

3. Have the authors made all data underlying the findings in their manuscript fully available?

Reviewer #1: Yes

Reviewer #2: No

Reviewer #3: Yes

4. Is the manuscript presented in an intelligible fashion and written in standard English?

Reviewer #1: Yes

Reviewer #2: Yes

Reviewer #3: Yes

Reviewer #1: This research evaluates route preview (RP) as a potential alternative to turn-by-turn (TBT) navigation for pedestrians. The findings indicate that while route preview may lack advanced features, it performs comparably to traditional navigation methods and is often preferred by users, especially in familiar environments. This suggests promising cognitive engagement benefits and positions route preview as a viable alternative. The research contributes valuable insights for data collection and fieldwork. However, careful attention to the article's structure, categorization of results, and presentation of graphs will enhance the clarity and impact of this work. Here are my detailed comments:

1. The references cited in this article on navigation technology are largely outdated. It is essential to incorporate recent studies to ensure that the text reflects current findings and developments in the field.

2. It seems that the text is not yet ready for submission for review. The numbering of subfigures has been forgotten. In the text, some figures are referred to with question marks instead of figure numbers.

3. A number of claims are made in the text without proof or evidence:

a. In the third paragraph of page 2, it is claimed that previous research has concluded that TBT instructions are not the best navigation option. This sentence requires several references.

b. In the second paragraph of page 3, it is claimed that RP has the potential to outperform TBT, which also requires rewriting the sentence and providing supporting evidence.

c. In the summary section on page 8, it is claimed that participants preferred RP more than the results indicate. The first sentence of Section 3.2.1 also states that only 43.7% of participants chose the RP method, which contradicts the previous claim. In this regard, the introduction to Section 4 should also be revised.

4. The structure of the article is not given in the introduction.

5. The title of Section 2 is the related works, but the features of pedestrian navigation are discussed in the preface and text.

6. It is better to also mention the survey number in the caption of the related tables.

7. Regarding Survey 3, the participation of only five individuals may be insufficient to draw generalizable conclusions. Consider discussing the implications of this small sample size for the findings.

Reviewer #2: The paper experimentally tested the relative benefit of TBT and RP navigation aid for pedestrian use. It reported three studies that elicits pedestrian preferences, compares human performance of using two alternatives, and suggests improvement on RP interface design. The study is interesting in adding to the debates between survey and route knowledge in navigation and relevant design of navigation aid. However, the paper has some critical flaws in framing research and experimental design.

(1) The paper lacks a clear articulation of research questions that address literature gaps. The pros and cons of survey and route knowledge in navigation were well established and I found little new in the framing of the problem. Without research questions, the three studies reported and their design could not be justified

(2) It is unclear why you need three studies to support your argument about RP vs TBT.

(3) The design if study #1 is flawed. The preferences of using TBT or RP are highly contextualized and dynamic, which makes it unsuitable to be discovered by online survey. There is no guarantee that people will report their real preferences that reflect what they actually do.

(4) The design of study #2 was also flawed. The TBT navigation instructions were not designed for learning and it is unfair to test their survey knowledge by asking them to sketch. The metrics for comparing the two modes were biased towards exploration and environment learning. There was also no explicit control of the level of familiarity of participants to the environment (University campus).

(5) The argument for doing Study #3 was not convincing. The work is best reported in a separate paper.

In conclusion, I found that the results of those studies were not sufficient to support their conclusion.

Reviewer #3: I have no major concerns or suggestions. I think this article is well-researched, well-written, and fascinating. In section 5.1, you might consider spelling out the date so no day/month vs. month/day confusion can take place. In a few places, there are figure links that have question marks instead of the figure numbers (I believe these are all related to figure 3). The formatting of the final three citations differs from that of the previous entries.

**Do you want your identity to be public for this peer review?** For information about this choice, including consent withdrawal, please see our Privacy Policy

Reviewer #1: No

Reviewer #2: **Yes:** Guoray Cai

Reviewer #3: No

---

## [Author Response · Author response to Decision Letter 1]

10 Nov 2025

RE: Response to Reviewers for Manuscript PONE-D-25-10260, "Evaluating Route Preview as an Alternative to Turn-By-Turn Navigation in Pedestrian Mobility"

Dear Dr Sehra,

We thank you and the reviewers for the insightful feedback and constructive criticism on our manuscript, "Evaluating Route Preview as an Alternative to Turn-By-Turn Navigation in Pedestrian Mobility" (PONE-D-25-10260). We appreciate the time and expertise you have dedicated to this review process. We have thoroughly revised the manuscript, taking every comment into careful consideration, and believe the paper is now substantially improved and suitable for publication in PLOS ONE.

We appreciate your editorial guidance. This feedback was pivotal, pushing us to articulate explicit research questions that now provide a robust theoretical scaffold for our three-study design.

Below, we provide a point-by-point response to the journal requirements and the specific concerns raised by each reviewer, detailing the changes made in the revised manuscript

Part 1: Response to Journal Requirements

We have carefully reviewed all journal-specific requirements to ensure the revised manuscript is fully compliant with PLOS ONE's formatting and publication standards. Below, we detail the actions taken for each requirement.

1. PLOS ONE Style Requirements: We have reviewed our manuscript against the provided PLOS ONE style templates and have updated the formatting, file naming, and overall structure to meet the journal's requirements. We are using the PLOS template version 3.6 Aug, August 2022, provided directly by Overleaf.

2. Inclusivity in Global Research Questionnaire: As requested, we have completed the PLOS questionnaire on inclusivity in global research and have uploaded the document as a Supporting Information file with our resubmission.

4. Financial Disclosure: We have updated our financial disclosure as requested. The statement now reads: "GS and JS - Swiss National Science Foundation (SNSF) grant number 200021 207430. The funders had no role in study design, data collection and analysis, decision to publish, or preparation of the manuscript."

5. Ethics Statement: The full ethics statement has now been integrated into the manuscript at the end of the Literature Review section (Section 2, page 6). The statement includes the full name of the approving committee ("Ethics Committee of the University of Monash") and the relevant project details ("Project Title: Exploring Navigation Skills using different methods, Project ID: 23126").

6. Copyrighted Map Images: We have addressed the issue of potentially copyrighted map images in the original Figures 1 and 4. The revised manuscript now only includes new, fully compliant illustrations. Therefore, we opted to remove Figure 1.

7. Figure 3 Reference: We have carefully checked all in-text figure references throughout the manuscript. All placeholder references (e.g., "???") have been replaced with the correct figure numbers, and we confirm that Figure 3 is now correctly referenced in the text.

8. Supporting Information Captions: Captions for all Supporting Information files have been added at the end of the manuscript, as per the journal's guidelines.

9. Identifying Data in Supporting Information: We have reviewed all Supporting Information files to ensure participant privacy. All potentially identifying data has been removed or fully anonymised. Please let us know if there is anything critical in the data set.

10. Consent for Interview Transcripts: We would like to clarify that our research did not involve the collection of interview transcripts. Study 3 was a design studio workshop, not a series of interviews. In accordance with our ethics approval, all participants provided written informed consent for their contributions (including anonymised quotes and design sketches) to be used for research purposes and publication.

Reviewer 1

We are grateful to Reviewer #1 for their valuable and constructive feedback. These comments have significantly improved the manuscript's structure, clarity, and engagement with the current state of the literature. Below, we address each point in detail.

1. References: We agree with the reviewer that the literature review needed updating. We have substantially revised the Literature Review (Section 2) to incorporate more recent and relevant studies, ensuring the manuscript reflects the current state of the field. As examples, we have added new citations such as Dugas et al. [5] and Mazurkiewicz et al. [6] that discuss contemporary challenges in pedestrian navigation.

2. Submission Readiness: We apologise for the formatting errors in the initial submission.

3. Unsupported Claims: We thank the reviewer for highlighting areas where our claims required stronger support.

◦ 3a. (Page 2 Claim) The claim that turn-by-turn (TBT) instructions are not the best navigation option for pedestrians has now been substantiated with multiple citations. In the revised manuscript (page 2), this statement is now supported by references [1, 5, 6, 12, 13].

◦ 3b. (Page 3 Claim) The statement regarding the route previews (RP) potential to outperform TBT has been rephrased to be more nuanced. It is now presented as a potential avenue for future design improvements that our study explores, rather than an unsubstantiated claim (page 3).

◦ 3c. (Contradictory Claims) We have revised the text to remove any contradictions. The summary in Section 3.2.4 (page 9) now accurately reflects the quantitative data presented in the results (a 43.7% preference for RP), stating that a "considerable share" of participants preferred RP, which avoids any overstatement.

4. Article Structure in Introduction: A new paragraph has been added at the end of the Introduction (page 3) that explicitly outlines the structure of the article, guiding the reader through the subsequent sections.

5. Section 2 Title: The title of Section 2 has been changed from "Related Works" to "Literature Review" to reflect its content more accurately. The introductory text of the section has also been revised accordingly.

6. Survey Number in Table Captions: We have updated all relevant table captions to include the corresponding study number for improved clarity (e.g., Table 1, Table 2, and Table 3).

7. Small Sample Size in Study 3: We acknowledge the reviewer's concern about the small sample size in Study 3. The Introduction to Study 3 (Section 5, page 15) has been revised to better clarify its qualitative, exploratory nature as a design studio workshop. Furthermore, we have added a paragraph to the Discussion (page 20) that explicitly acknowledges the implications of this small sample size as a limitation of the study.

We believe these changes have greatly strengthened the manuscript and thank the reviewer again for their guidance.

Reviewer 2

We are particularly grateful to Reviewer #2 for their incisive critique. These comments challenged us to re-evaluate the manuscript's core narrative, prompting a significant reframing that has substantially elevated its scientific contribution

1. Lack of Clear Research Questions: We agree that research questions help to better guide readers. We have formulated three explicit Research Questions (RQ1, RQ2, RQ3), which are now presented in the Introduction (page 2). These questions frame the entire study by clarifying the specific literature gaps we aim to address concerning pedestrian preferences, objective performance, and design enhancements for navigation aids.

2. Justification for Three Studies: The newly introduced Research Questions provide a clear and logical justification for the three-stage study design. The manuscript now explicitly links each study to a specific research question, creating a cohesive narrative:

◦ Study 1 addresses RQ1 by investigating the stated preferences of pedestrians.

◦ Study 2 addresses RQ2 by evaluating the objective performance of the two navigation modes.

◦ Study 3 addresses RQ3 by exploring potential design enhancements for RP.

3. Flawed Design of Study #1: We acknowledge the inherent limitations of using online surveys for assessing contextual preferences. In the revised manuscript, we have added a new paragraph in Section 3 (page 6) that explicitly recognises this limitation. The text now frames Study 1 as an essential first step that provides a baseline understanding of user attitudes, which is then tested against observed behaviour in Study 2.

4. Flawed Design of Study #2: We have added a comprehensive new paragraph in the methodology section for Study 2 (Section 4.1.1, page 10) to justify our design choices more robustly. This section now explains:

◦ The rationale for comparing TBT and RP on spatial learning tasks, which reflects a realistic user need to recall routes even when using a navigation aid.

◦ The controls we implemented to manage participant familiarity with the environment. This includes the recruitment of newly enrolled students and the use of familiarity as a covariate in our regression analysis, which confirmed its expected effect (Table 3).

◦ How these additions provide a more rigorous and balanced methodological framework for the field study, strengthening the validity of our findings. These additions not only resolve the reviewer's specific methodological concerns but also provide a more rigorous framework that strengthens our central—and perhaps surprising—finding: that a feature-limited RP mode performs on par with a fully-featured TBT system, even under more stringently controlled conditions.

5. Argument for Study #3: The Introduction to Study 3 (Section 5, page 15) has been completely rewritten. It is now clearly framed as the logical next step that directly follows from the findings of Studies 1 and 2. It addresses the key design question that emerges from our results: how could RP be improved to outperform TBT potentially?

We are confident that these substantial revisions have addressed the reviewer's critical concerns and have resulted in a significantly stronger and more coherent manuscript. Thank you once again for your time and effort.

Reviewer 3

We thank Reviewer #3 for their careful reading and helpful suggestions, which have improved the manuscript's professionalism and readability.

1. Date Formatting: We have corrected the date formatting in Section 5.1. The date is now spelt out as "June 12, 2024" (page 15) to prevent any ambiguity.

2. Figure Link Placeholders: The entire manuscript has been reviewed, and all placeholder figure references have been corrected with the proper figure numbers.

3. Citation Formatting: The reference list has been carefully checked, and the formatting of all citations has been standardised for consistency throughout the list.

Conclusion

We once again thank the editor and all reviewers for the time and effort invested in providing this constructive feedback. We are confident that the extensively revised manuscript is now a much stronger scientific contribution and is suitable for publication in PLOS ONE. We look forward to your evaluation of the revised submission.

Kind regards,

Gian-Luca Savino, Emanuel de Bellis, Reuben Kirkham, Johannes Schöning (Corresponding Author)

---

## [Decision Letter · Decision Letter 1]

26 Dec 2025

Evaluating Route Preview as an Alternative to Turn-By-Turn Navigation in Pedestrian Mobility

PONE-D-25-10260R1

Dear Dr. Schöning,

We’re pleased to inform you that your manuscript has been judged scientifically suitable for publication and will be formally accepted for publication once it meets all outstanding technical requirements.

Kind regards,

Sukhjit Singh Sehra

Academic Editor

PLOS One

Additional Editor Comments (optional):

Congratulations, you have addressed all the comments of the reviewers, the manuscript is ready for the next steps.

Reviewers' comments:

Reviewer's Responses to Questions

**Comments to the Author**

Reviewer #1: All comments have been addressed

Reviewer #3: All comments have been addressed

2. Is the manuscript technically sound, and do the data support the conclusions?

Reviewer #1: Yes

Reviewer #3: Yes

3. Has the statistical analysis been performed appropriately and rigorously?

Reviewer #1: Yes

Reviewer #3: Yes

4. Have the authors made all data underlying the findings in their manuscript fully available?

Reviewer #1: Yes

Reviewer #3: Yes

5. Is the manuscript presented in an intelligible fashion and written in standard English?

Reviewer #1: Yes

Reviewer #3: Yes

Reviewer #1: (No Response)

Reviewer #3: I think these edits address mine and the other the reviewers' concerns well, and make the paper much stronger and more clear to read.

**Do you want your identity to be public for this peer review?** For information about this choice, including consent withdrawal, please see our Privacy Policy

Reviewer #1: No

Reviewer #3: No

---

## [Editor Report · Acceptance letter]

PONE-D-25-10260R1

PLOS One

Dear Dr. Schöning,

I'm pleased to inform you that your manuscript has been deemed suitable for publication in PLOS One. Congratulations! Your manuscript is now being handed over to our production team.

Kind regards,

on behalf of

Dr. Sukhjit Singh Sehra

Academic Editor

PLOS One